# Genetic Markers Regulating Blood Pressure in Extreme Discordant Sib Pairs

**DOI:** 10.3390/genes14101862

**Published:** 2023-09-25

**Authors:** Kevin M. O’Shaughnessy

**Affiliations:** Experimental Medicine & Immunotherapeutics Division, Department of Medicine, University of Cambridge, Cambridge CB2 0QQ, UK

**Keywords:** blood pressure, microsatellite markers, chromosomes 11 and 17, extreme discordant sib pairs

## Abstract

Genome-wide scans performed in affected sib pairs have revealed small and often inconsistent clues to the loci responsible for the inherited components of hypertension. Since blood pressure is a quantitative trait regulated by many loci, two siblings at opposite extremes of the blood pressure distribution are more likely to have inherited different alleles at any given locus. Hence, we investigated an extreme discordant sib pair strategy to analyse markers from two previous loci of interest: (1) the Gordons syndrome locus that includes the WNK4 gene and (2) the ROMK locus identified in our first genome-wide scan. For this study, 24 sib pairs with strong family histories of essential hypertension were selected from the top and bottom 10% of the blood pressure distribution and genotyped for highly polymorphic microsatellite markers on chromosomes 11 and 17. The mean age of the population was 39.8 ± 7.8 years. A significant inverse correlation was found between the squared difference in pulse pressure and the number of alleles shared by IBD between the siblings for the DS11925 marker (r = −0.44, *p* = 0.031), systolic pressure and chromosome 17 markers (D17S250: r = −0.42, *p* = 0.040; D17S799 (r = −0.51, *p* = 0.011), and this relationship persisted after correcting for age and gender. Markers on chromosome 17 (D17S250, D17S928 and D17S1301) and 11 (D11S1999) also correlated with diastolic pressure. These results illustrate the successful use of discordant sib pair analysis to detect linkage within relatively small numbers of pedigrees with hypertension. Further analysis of this cohort may be valuable in complementing findings from the large genome wide scans in affected sib pairs.

## 1. Introduction

High blood pressure is a common condition affecting one in four adults in the UK, and it is an important modifiable risk factor for cardiovascular, cerebrovascular and renal diseases globally [1]. Genetic and environmental factors impact variations in blood pressure amongst individuals; however, the mechanisms underlying this regulation are poorly understood. A number of studies using a genetic linkage approach identified several candidate genes influencing the blood pressure level, but dissecting these major genes is problematic due to blood pressure’s complex, polygenic nature. Since the advent of the genome era, a number of genome-wide scans (GWS) performed in hundreds of affected sib pairs (ASPs) [2,3] and in over hundred thousand individuals [4,5,6,7,8], including the UK Biobank sample cohort [9], have not revealed consistent clues to the loci responsible for hypertension. The reported genome-wide association studies (GWAS) identified over 1000 loci influencing blood pressure (BP), but the variants were generally limited to common and low frequencies that are mostly in the intergenic region of the genome, showing small associations with BP. On the other hand, several exome-sequencing and chip studies identified multiple rare variants in coding regions with large associations [10,11,12]. The most recent large-scale whole-genome-sequencing meta-analysis of blood pressure phenotypes (systolic, diastolic blood pressure and hypertension) in several multi-ancestry cohorts (51,456 participants) identified only two signals (rare and common variants) that achieved genome-wide significance [9]. This confirms the widely assumed genetically complex and polygenic nature of blood pressure.

In the mid-1990s, one approach received particular attention for dissecting polygenic traits including obesity, asthma, and age-related macular degeneration [13,14,15], and this is the extreme discordant sib pair (EDSP) strategy proposed by Risch and Zhang [16,17]. Here, individuals are sampled from both ends of the phenotype distribution, and because they share the same genetic background, EDSPs provide the most informative siblings for detecting genetic linkage. Using the extreme threshold strategy, Risch and Zhang showed that one can increase the power of a pair, and they estimated the number of required sib pairs to be as few as 10/25 at the two ends of the scale compared to the many hundreds required for the affected sib pairs needed for a linkage analysis [10,16,18] or the several hundreds of thousands required for GWAS [19,20]. We tested the hypothesis that if a genetic locus contributes to blood pressure regulation, then two siblings at opposite extremes of the blood pressure distribution will inherit different alleles at the locus.

Our aim was to recruit the untreated offspring of affected sibling pairs with essential hypertension from the East Anglia Region community [21] and determine what the detection rate would be for extreme discordant sib pairs in this cohort. We applied the EDSP strategy to the untreated offspring of hypertensive parents to validate two previous loci associated with hypertension: (i) the Gordon’s syndrome (psuedohypoaldosteronism type II, PHA type 2) locus on chromosome 17, including the *WNK4* gene [22,23], and (ii) the *ROMK* locus on chromosome 11 which was identified in our first GWS for hypertension [21]. The two loci were selected from our published work in affected sib pairs [21] due to ongoing interest in the monogenic and polygenic nature of hypertension, in particular Gordon’s syndrome and *WNK* kinases [22,24,25].

## 2. Material and Methods

In total, 140 families with two or more untreated siblings with a family history of essential hypertension were recruited, of which 47 offspring pairs, referred to herein as sib pairs, had blood pressure (BP) differences of greater than 15/14 mmHg. Of the 47 sib pairs, 24 sib pairs (48 individuals) fitted the strictest definition of extreme discordance. Establishing the genetic architecture in individuals with extreme trait values is important because these individuals are at an increased risk of disease and are also the most likely to harbour rare variants of large effects due to natural selection. In these sib pairs, either one sib had high blood pressure (SBP >139 mmHg or DBP >85 mmHg) and the other sib had low blood pressure (SBP <112 mmHg or DBP <66 mmHg) or there was a difference of at least >25/20 mmHg in blood pressure between the sibs. Centiles of BP were defined from reference data previously gathered from 35,000 healthy subjects registered at their general practitioners and recruited to for cardiovascular risk screenings in the CLEAREST (Cholesterol Levels in the East Anglia Region Effective Screening and Treatment) study [26]. Ethical approval was obtained from the Local Research Ethics Committee (LREC 93/8), and written informed consent obtained from each participant.

At each participant’s clinic visit, basic demographic information (age, sex, self-reported birthweight in pounds, smoking status, alcohol consumption, personal illnesses and treatments) was acquired and a family history questionnaire was completed. Height and weight were recorded using standard methods in kilograms, and the body mass index was calculated in kg/m^2^. Peripheral blood pressure was recorded in the brachial artery of the non-dominant arm using a validated oscillometric technique (HEM-705CP; Omron Corporation, Kyoto, Japan). Blood pressure was measured twice unless there was discrepancy in the two measurements (>10 mmHg); in this case, a third reading was taken, and the average value was used in the subsequent analysis. All the sib pairs were healthy and were not taking any hypertensive or cardiovascular drugs. Peripheral venous blood samples were obtained for biochemical and genetic analysis.

Genomic DNA (gDNA) was isolated from whole blood in all 47 sib pairs recruited, but only individuals who fitted the strictest BP criteria i.e., 24 sib pairs, were genotyped for highly polymorphic microsatellite markers (all CA repeats) from the MRC panel and Gordon’s markers located on chromosomes 11 and 17, as listed in Table 1 [21,27]. Using previously investigated fluorescent-labelled primers and published methods, a polymerase chain reaction (PCR) carried out with minor modifications [21]. Specifically, microsatellite loci were amplified via the PCR in a 10μL reaction volume containing 10mM of Tris/HCl (at a pH of 8.3), 50 mM of KCl, and 0.2 mM of each dNTP, 2 pmol of each primer (forward and backward), an optimal concentration of MgCl_2_ (from 1.5 to 2.0mM; determined separately for each primer pair) and 50–100 ng of template gDNA. The polymerase chain reactions were carried out for 34 cycles at 94 °C for three minutes, 55 °C for 1 min and 72 °C for 45 s. The fluorescent-labelled PCR products were pooled and loaded onto an ABI 377 semi-automated sequencer (Perkin Elmer). The sizes of the markers, which reflected differences in the number of CA repeats, were assigned using version 2 of ABI Genotyper^®^ software and corrected manually if necessary. To identify if any genetic polymorphisms in the *WNK4* gene exons were involved in this cohort (n = 24 sib pairs), a mutation analysis of exons 7 and 17 was performed via a single-strand conformation polymorphism (SSCP) analysis and direct sequencing on an ABI 377 sequencer, and the data were analysed using Genescan and Genotyper software from ABI (Perkin Elmer).

A statistical analysis was performed using SPSS software (version 28.0) to determine the relationship between blood pressure and the microsatellite markers (correlation and multiple regression analyses) and for graphical illustration. The data were also analysed using the Haseman–Elston quantitative trait loci regression method in which the squared difference in the phenotype (here, blood pressure) between sibs is regressed to the number of alleles shared by identity-by-descent (IBD) and by the use of the SPLINK package, a non-parametric likelihood ratio method. Since this was a small study of 24 sib pairs, correlation coefficients and unadjusted p-values are presented, and a *p*-value of <0.05 was considered statistically significant.

## 3. Results

The demographic details of this cohort are provided in Table 2. The mean age of the population was 39.8 ± 7.8 years (an age range 28–67 years), and the average blood pressure was 125 ± 18/77 ± 13 mmHg (Table 2). The two discordant groups of siblings were not significantly different in terms of their age, average height, birthweight or serum markers but differed in their body weight and body mass index. As expected, the affected siblings (high-blood-pressure tail) had significantly higher systolic, diastolic and pulse pressures compared to their unaffected siblings (low-blood-pressure tail). No significant differences were found between the two groups for any of the serum markers.

Blood pressure levels correlated with chromosome 17 and 11 microsatellite markers, suggesting evidence of linkage. Significant inverse correlations were identified between the squared difference in the systolic blood pressure and the number of alleles shared between the siblings for the D17S799 (r = −0.51, *p* = 0.011) and D17S250 (r = −0.42, *p* = 0.04) markers, as illustrated in Figure 1. Diastolic blood pressure also correlated with markers on chromosome 17, D17S1301 (r = −0.42, *p* = 0.04; Figure 2), D17S250 (r = −0.39, *p* = 0.05; Figure 2) and D17S928 (r = 0.42, *p* = 0.04, Figure 3), and the D11S1999 marker on chromosome 11 (r = −0.46, *p* = 0.025; Figure 3). These relationships persisted even after correcting for confounding factors including age and sex for the markers associated with systolic and diastolic pressure (Table 3). A significant inverse correlation was also observed between the DS11925 marker and the squared difference in pulse pressure (Figure 4, r = −0.44, *p* = 0.031), a surrogate marker of arterial compliance and an independent predictor of cardiovascular complications.

However, we did not find any polymorphisms in exons 7 and 17 of the *WNK4* gene in this cohort, suggesting that within the population from which sib pairs were drawn (48 individuals genotyped), there were no common polymorphisms in the coding region of the *WNK4* gene that influenced the blood pressure level.

## 4. Discussion

High blood pressure affects over 25% of adults and is the major cause of cardiovascular morbidity and mortality. Despite over half a century of multiple lines of investigations, the genetic basis remains poorly understood, with each genetic loci contributing to a minor proportion of blood pressure inheritance. We investigated the feasibility of exploiting the continuous nature of the blood pressure distribution, using extreme discordant sib pairs rather than regarding hypertension as a categorical abnormality. This strategy is widely used in experimental hypertension genetics in which two rodent strains of opposite phenotypes are crossed. However, this approach is most useful when there are large differences in the continuous variable between sib pairs. The extreme discordant strategy, in which a lack of alleles at contributing loci is sought, is predicted to reduce the number of sib ships required by several orders of magnitude (from 10- to 40-fold) [16,17] but depends on identifying families in which the candidate gene is sufficiently important for extreme discordance in blood pressure to exist between sibs with different alleles. The main impetus for this approach is a practical one: it reduces the effort, time and genotyping costs versus a whole-genome scan.

For the first time, we were able to identify that extreme discordant sib pairs do indeed exist in the general population, although their identification is even more labour intensive clinically because of the low return for each proband when compared to that of the concordant sibs required for affected sib pair studies. In theory, our present cohort approached the number required in humans, when the number of loci responsible for hypertension is itself unknown. We were able to recruit double the minimum target proposed by Risch. However, the identification and phenotyping process carried out to identify 24 sib pairs with extreme blood pressure differences took 2 years. Our East Anglian population also has a low prevalence of cardiovascular risk compared to London and other parts of the UK. All the sib pairs studied, including the original probands (parents), were local residents from a region of low CV risk with little or no influence of environmental factors. Nevertheless, this resource will be useful to replicate loci of modest linkage on genome-wide scans and will help to exclude false positives.

In our previously published genome-wide search for the susceptibility loci of hypertension in affected sib pairs (ASPs) [21], we identified microsatellite markers on a chromosome 11q locus that associated with hypertension. The D11S934 marker showed significant two-point linkage (*p* = 0.004) in the affected sib-pairs [21]. This region contains the *ROMK* gene that encodes the *KCNJ1* potassium channel and is mutated in Bartter’s syndrome (type 2), a monogenic hypertension disorder [28]. In our study here, the D11S934 marker did not associate with any blood pressure phenotype (systolic, diastolic or pulse pressure). It is possible *ROMK* does not influence blood pressure in our East Anglian population, although other genes in the region around D11S934 may be operating. In fact, we found several other highly polymorphic genomic markers in the chromosome 11 region associated with diastolic pressure (D11S1999, D11S1301 and D11S928) and pulse pressure (D11S925) phenotypes.

Similarly, two of the markers investigated for chromosome 17 correlated with systolic blood pressure (D17S799 and D17S250) suggesting genetic linkage. We previously demonstrated evidence of linkage for the D17S250 marker (with a maximum LOD score of around 2.3) with Gordon syndrome in a large, well-characterised Brisbane family with seven offspring [22]. This suggests that the same genetic locus for hypertension is involved in both “Gordon syndrome” and “essential hypertension”. So far, there is no evidence to corroborate our findings with this EDSP strategy in other populations or other complex phenotypes using microsatellite markers. Numerous studies sought to identify genes for hypertension and blood pressure using genome-wide scans via discordant sib pairs and general populations. But using the genome scan, the only study that reported regions containing five genomic markers on various chromosomes (D3S2387, D11S2019, D15S657, D16S3396 and D17S1303) for systolic and diastolic blood pressure in their primary analysis and eight regions in a secondary analysis (D4S3248, D72195, D101423, D20S470, D20S482, D21S2052, PAH and AGT) highlighted chromosomes 11 and 17 as putative regions for harbouring hypertension genes [29]. This study used extremely discordant, highly concordant and low concordant sib pairs in China [29], unlike the American, Quebec and San Antonio studies [30,31,32].

## 5. Conclusions

Significant inverse correlations between blood pressure (systolic, diastolic and pulse pressures) and several microsatellite markers suggests evidence of linkage on chromosomes 11 and 17, as noted previously in our studies on affected sib pairs with essential hypertension and the Brisbane family with Gordon’s syndrome. The results also illustrate the use of a discordant sib pair analysis to detect linkage within small numbers of pedigrees with essential hypertension. Further analysis of this cohort may be valuable in complementing findings from large genome-wide scans in affected sib pairs.

## Figures and Tables

**Figure 1 genes-14-01862-f001:**
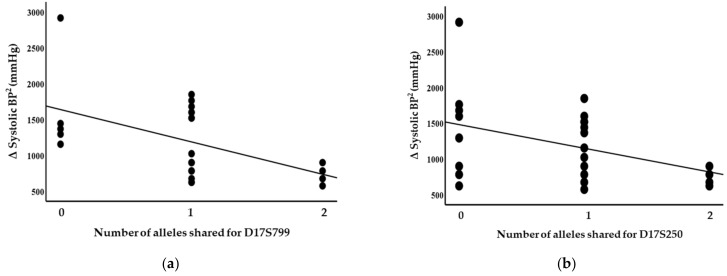
Systolic blood pressure associated with chromosome 17 markers in siblings sharing alleles at IBD, suggesting evidence of linkages with (**a**) D17S799 (*p* = 0.011) and (**b**) D17S250 (*p* = 0.04).

**Figure 2 genes-14-01862-f002:**
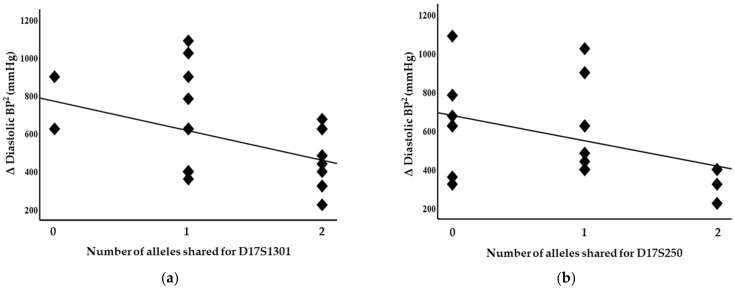
Diastolic blood pressure associated with chromosome 17 markers in siblings sharing alleles at IBD, suggesting evidence of linkages with (**a**) D17S1301 (*p* = 0.04) and (**b**) D17S250 (*p* = 0.05).

**Figure 3 genes-14-01862-f003:**
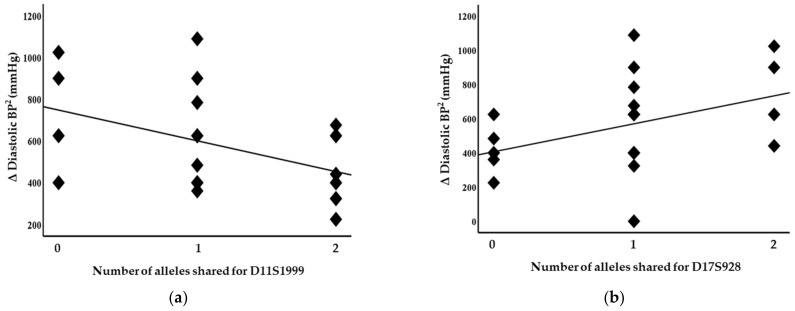
Diastolic blood pressure associated with chromosome 11 and 17 markers in siblings sharing alleles at IBD, suggesting evidence of linkages with (**a**) D11S1999 (*p* = 0.025) and (**b**) D17S928 (*p* = 0.04).

**Figure 4 genes-14-01862-f004:**
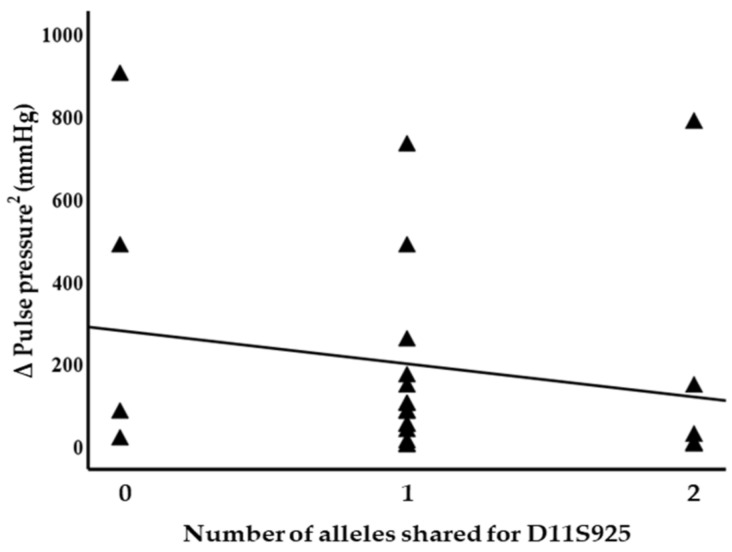
Pulse pressure associated with chromosome 11 marker in siblings sharing alleles at IBD, suggesting evidence of linkage with D11S925 (*p* = 0.03).

**Table 1 genes-14-01862-t001:** DNA microsatellite markers investigated on chromosomes 11 and 17.

Chromosome 11			Chromosome 17		
Locus	Marker	Range (bp)	Locus	Marker	Range (bp)
D11S910	AFM154yh2	249–261	D17S250	Mfd15	151–169
D11S912	AFM157xh6	101–123	D17S787	AFM095tc5	138–166
D11S925	AFM220yb6	173–199	D17S789	AFM107yb8	154–170
D11S934	AFM248wf5	230–	D17S793	AFM165zd4	95–109
D11S968	AFM109xc3	137–155	D17S798	AFM179xg11	218–
D11S1984	GGAA17G05	166–206	D17S799	AFM192yh2	186–200
D11S1392	GATA6B09	200–220	D17S807	AFM234xc9	114–138
D11S1999	GATA23F06	109–137	D17S809	AFM157xh6	229–247
D11S2000	GATA28D01	199–235	D17S928	AFM217yd10	135–165
D11S2371	GATA90D07	193–213	D17S934	-	174–
D11S4464	GATA64D03	225–249	D17S1290	GATA49C09	170–210
			D17S1293	GGAA7D11	262–290
			D17S1301	GATA28D11	147–163
			D17S1303	GATA64B04	225–245
			D17S1308	GATAT1A05	304–316

**Table 2 genes-14-01862-t002:** Demographic, haemodynamic and biochemical data in sib pairs.

Variables	Sib Pairs			
	All (Mean ± SD)	Affected * (Mean ± SD)	Unaffected * (Mean ± SD)	Significance (*p* =)
Age (years)	39.8 ± 7.8	40.7 ± 7.3	38.8 ± 8.4	*ns*
Gender (Male/Female)	15/33	10/14	5/19	-
Height (m)	1.66 ± 0.1	1.67 ± 0.1	1.65 ± 0.1	*ns*
Weight (kg)	71.8 ± 14.2	77.0 ± 15.2	66.3 ± 11.4	0.009
Body mass index (kg/m^2^)	26.1 ± 4.9	27.7 ± 5.4	24.3 ± 3.8	0.018
Birthweight (lbs)	6.80 ± 1.3	6.92 ± 1.2	6.64 ± 1.4	*ns*
Systolic blood pressure (mmHg)	125 ± 18	143 ± 7	108 ± 5	0.001
Diastolic blood pressure (mmHg)	77 ± 13	89 ± 5	66 ± 5	0.001
Pulse pressure (mmHg)	48 ± 8	52 ± 8	43 ± 6	0.001
Total cholesterol (mmol/L)	5.24 ± 0.8	5.23 ± 0.8	5.24 ± 0.8	*ns*
Glucose (mmol/L)	141 ± 2.5	142 ± 2.5	141 ± 2.6	*ns*
Sodium (mmol/L)	4.61 ± 0.1	4.76 ± 0.3	4.46 ± 0.5	*ns*
Potassium (mmol/L)	5.22 ± 0.5	5.19 ± 0.6	5.24 ± 0.4	*ns*
Urea (mmol/L)	4.8 ± 0.3	4.3 ± 1.0	5.3 ± 1.6	*ns*
Creatinine (mmol/L)	80.7 ± 3.6	75.8 ± 14.6	85.6 ± 15.6	*ns*

*ns*—Not significant; * Top and bottom 10% of blood pressure distribution.

**Table 3 genes-14-01862-t003:** Genetic markers independently associated with blood pressure phenotypes.

Model Parameters	Beta	Significance Level (*p*)
Dependent variable: systolic blood pressure		
Age	–0.027	0.913
Sex	–0.096	0.639
D17S799	–0.545	0.030
*Adjusted R^2^ value = 0.272*		
Age	0.173	0.438
Sex	–0.084	0.694
D17S250	–0.379	0.084
*Adjusted R^2^ value = 0.204*		
**Dependent variable: diastolic blood pressure**		
Age	0.095	0.664
Sex	–0.168	0.429
D11S1999	–0.429	0.048
*Adjusted R^2^ value = 0.233*		
Age	0.110	0.624
Sex	–0.208	0.338
D17S250	–0.378	0.086
*Adjusted R^2^ value = 0.195*		
Age	0.199	0.353
Sex	–0.155	0.470
D17S1301	–0.394	0.062
*Adjusted R^2^ value = 0.216*		
Age	0.412	0.040
Sex	–0.292	0.130
D17S928	0.482	0.015
*Adjusted R^2^ value = 0.363*		
**Dependent variable: pulse pressure**		
Age	0.007	0.975
Sex	0.386	0.106
D11S925	–0.315	0.161
*Adjusted R^2^ value = 0.168*		

## Data Availability

No new data were created or analyzed in this study. Data sharing is not applicable to this article.

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
