# Peer review of "Genetic Markers Regulating Blood Pressure in Extreme Discordant Sib Pairs"

_genes, 2023, doi:10.3390/genes14101862_

Round 1
Reviewer 1 Report (New Reviewer)
The article by Yasmin and K.M. O’Shaughnessy analyzes markers from two loci. However, as it stands, this paper is relatively superficial, both conceptually and technically. The original hypothesis is not clear in the present study. What is the novelty of present study?
The introduction is confusing.
What is the criterion for the selection of these two loci?
Based on which parameters was the research conducted only in the local East of England community?
Why do authors investigate sib with low blood pressure? Please explain.
In the material and methods, the authors say that they use different demographic parameters. Why is birth-weight important?
The authors didn’t discuss the results presented in Table 2. What is the link between blood pressure levels and biochemical parameters? Why are these results important?
Literature must be upgraded. Only two references were written less than 5 years ago, nothing new was used.
Author Response
We would like to respectfully pick up on what the reviewers have said here. Please see our point-by-point responses below.
Comments and Suggestions for Authors
1. The article by Yasmin and K.M. O’Shaughnessy analyzes markers from two loci. However, as it stands, this paper is relatively superficial, both conceptually and technically. The original hypothesis is not clear in the present study. What is the novelty of present study?
We are highly disappointed with reviewer 1 and respectfully disagree on his above comment, as reviewer 2 and a previous reviewer indicated this, as an eloquent novel study which employed a pioneering approach of analysing genetic markers for blood pressure, a complex polygenetic and environmental condition. The reviewers also stated study findings have tangible and translatable implications for future studies generating interest to genetics readership and the study provides advancement in the field.
The previous reviewers also commented that the study aim is clearly stated, manuscript well written in a clear and concise manner, data is well interrogated, interpreted and presented.
It is therefore disappointing to learn reviewer 1 indicated it as a superficial study, both conceptually and technically!
We have amended the study hypothesis for clarity in revised manuscript as it was not clear for the reviewer.
The use of extremely discordant sib pairs is particularly uncommon because of the difficulty and expense in accruing such samples and the screening itself is laborious. For this reason alone, we have seen very few genetic linkage studies that have used this strategy and published in human complex disorders such as blood pressure, when measured on a continuous scale. As mentioned in the discussion it took us 2 years to actually find the 24 extremely discordant sib pairs used for this study. So, we are unclear on what the reviewer expected from a small study like this.
The novelty of this study is its study design, the EDSPs approach used in selecting healthy sib pairs at the two tails of the blood pressure scale who were never diagnosed or treated for blood pressure, but had a strong family history of essential hypertension (i.e., affected sib pairs previously identified for GWS, see Ref. 21). To this extent, our sib pairs are enriched for hypertension susceptibility genes. Additionally, the subjects were local residents originating from a region of low cardiovascular risk compared to other parts of UK.
2. The introduction is confusing.
We respectfully disagree with this comment, as other reviewers made positive comments (see our above response). However where possible we have tried to simplify the background and provided additional references in the introduction. Hence all the reference numbers have changed in the revised manuscript.
3. What is the criterion for the selection of these two loci?
The two loci and panel of genetic markers tested were specifically selected based on our own interest in these loci (we have published GWS in affected sib pairs and Gordon syndrome families) and other extensive literature of the loci suggesting direct involvement in essential hypertension.
4. Based on which parameters was the research conducted only in the local East of England community?
The local community population we refer to in the introduction are the affected sibling pairs (ASPs) who previously participated in a GW Search for loci linked to essential hypertension is from the East Anglia Region. More details on this group are published in new reference 21.
5. Why do authors investigate sib with low blood pressure? Please explain.
We are not using just low BP individuals but discordant sibs at the ends of the BP distribution. The idea of using extreme discordant sib pairs from the opposite ends of the blood pressure scale is to identify genes causing hypertension in the family, since two siblings are likely to have inherited different copies of such genes.
6. In the material and methods, the authors say that they use different demographic parameters. Why is birth-weight important?
We are unclear on this comment.
To clarify on the above comment on using different demographic parameters, as with any research study, as a standard procedure any researcher would collect not just age, sex, height and weight but also information relating to participants personal habits, health status and family history and also birth weight. We basically reported all the information that was collected and data recorded at the screening visit in manuscript text and Table 2. Therefore, we did not make any changes in the revised manuscript, as birth weight was not the focus of our study, but provided birthweight importance in cardiovascular disease (CVD) below.
Birthweight we all know is closely related with adult body mass index, and is a critical factor in the development of cardiovascular disease (CVD). David Barker postulated the foetal onset adult disease or Thrifty phenotype on low birthweight association with hypertension and this is evidenced by many studies. Additionally, both low birth weight and obesity in adulthood has also been shown to stimulate the sympathetic nervous system and alter renal function, thereby increasing the risk of CVD. Again this link was not our study objective.
7. The authors didn’t discuss the results presented in Table 2. What is the link between blood pressure levels and biochemical parameters? Why are these results important?
To note, in the original manuscript submitted to Genes, as a customary practise only demographic, anthropometric and phenotype data, i.e., blood pressure relevant to the study was presented in Table 2. However, since a previous reviewer asked for data on blood markers, we merely presented this information in the revised Table 2. To reiterate, our study focus was identifying EDSPs and validating previous loci associated with hypertension, but not investigating links or associations between blood pressure levels and biochemical markers.
Moreover, we performed this study in healthy offspring of ASPs, so we did not expect to see significant group differences for these serum markers, which are routinely measured in the clinics. To reiterate, our aim was on genetic markers influencing blood pressure and we therefore, did not accentuate this findings either in the results or discussion. Nonetheless, we have incorporated a sentence on these results in revised manuscript text (last sentence in the 1st paragraph).
8. Literature must be upgraded. Only two references were written less than 5 years ago, nothing new was used.
Please see our response above in point 2.

Reviewer 2 Report (New Reviewer)
Review for the paper “Genetic markers regulating blood pressure in extreme discordant sib pairs”
The authors wanted to investigate two loci of interest, the Gordons syndrome locus that includes the WNK4 gene, and the ROMK locus associated with hypertension. They investigated 24 sib pairs with a family history of essential hypertension. They found that the DS11925 marker is correlated with systolic blood pressure and also, the D17S250, D17S928, D17S1301 markers located on chromosome 17 and the D11S1999 marker located on chromosome 11 were correlated with diastolic blood pressure.
Comments
1. The introduction provides background relevant for this study.
2. The study provides an advance in the field.
3. The objectives and the rationale of the study are clearly stated.
4. Please provide the number of the ethics commission's opinion.
5. The methods are well described. The authors said that the sequences of the primers were mentioned in a previous study. Please provide the reference or provide again the sequences of the primers user for PCR reaction.
6. The interpretation of results and study conclusions supported by the data.
Author Response
We would like to respectfully pick up on what the reviewers have said here. Please see our point-by-point responses below.
Comments and Suggestions for Authors
Review for the paper “Genetic markers regulating blood pressure in extreme discordant sib pairs”
The authors wanted to investigate two loci of interest, the Gordons syndrome locus that includes the WNK4 gene, and the ROMK locus associated with hypertension. They investigated 24 sib pairs with a family history of essential hypertension. They found that the DS11925 marker is correlated with systolic blood pressure and also, the D17S250, D17S928, D17S1301 markers located on chromosome 17 and the D11S1999 marker located on chromosome 11 were correlated with diastolic blood pressure.
We thank reviewer 2 for their positive and constructive comments on clarity of the introduction, study rationale, methodology, findings and conclusions.
Comments
1. The introduction provides background relevant for this study.
-
2. The study provides an advance in the field.
-
3. The objectives and the rationale of the study are clearly stated.
-
4. Please provide the number of the ethics commission's opinion.
We apologise for this oversight. In the revised manuscript we have provided the Ethics committee reference number (LREC 93/8) as requested.
5. The methods are well described. The authors said that the sequences of the primers were mentioned in a previous study. Please provide the reference or provide again the sequences of the primers user for PCR reaction.
We thank the reviewer for his positive comment on the methodology.
We actually provided previous study reference which was number 17 (reference provided below) in the manuscript introduction and methods sections.
However, this number has changed to reference 21 in the revised manuscript.
- Sharma, P.; Fatibene, J.; Ferraro, F.; Jia, H.; Monteith, S.; Brown, C.; Clayton, D.; O’Shaughnessy, K.M.; Brown, M.J. A genome-wide search for susceptible loci to human essential hypertension. Hyperten. 2000, 35, 1291-1296.
6. The interpretation of results and study conclusions supported by the data.
We thank the reviewer for his positive comment on our study findings and conclusions.

Round 2
Reviewer 1 Report (New Reviewer)
Authors well addressed my previous comments. The paper improved very much
This manuscript is a resubmission of an earlier submission. The following is a list of the peer review reports and author responses from that submission.
Round 1
Reviewer 1 Report
Yasmin and O’Shaughnessy used a discordant sib pair approach to assess associations of genetic variants with blood pressure traits at two loci: WNKR (Gordons syndrome locus) and ROMK.
Major comments:
- The investigation of only 2 loci needs to be explained more. I do not understand why the analysis was restricted to just two loci. In the current manuscript, this decision seems to be arbitrary. Surely, it would be more comprehensive to assess all assessed genetic variants. Indeed, hypothesis-free (i.e. not candidate gene) studies have proven to be more robust to detecting true positives (i.e. results that can be replicated). I have strong reservations about the choice of selecting only 2 loci without much of an explanation, and I have concerns that the results deemed significant by the authors are just borderline significant by chance. In addition, I am assuming the authors used p<0.05 to denote significant, and that multiple testing was not accounted for. Again, this setting is not suitable to reliably detect true associations.
- With regard to the description of the cohort and the demographic table. It is written that gender is assessed. Can the authors confirm that indeed they mean self-reported gender as a social construct or are they perhaps referring to genetic sex? I usually see genetic association studies adjusting models for sex rather than gender. It is essential to use the correct terminology to avoid confusion so that is why I am asking for clarification on this point.
- All sections of the manuscript (from the introduction to conclusion) were very brief and lacked sufficient detail in my opinion.
Minor comments:
- Human gene names should be italicized.
- When providing the mean in the text, also include the corresponding standard deviation, as what is done in the demographic table (e.g. for the mean age of the population provided in the abstract).
- For the demographics table, I am curious as to why Birthweight is in pounds, but the other weights are in Kg. Is there a reason for this change in units?
- Some formatting issues (e.g. in the last sentence of the conclusion there seems to be an space before the final e in the word large)
Author Response
We would like to respectfully pick up on what the reviewers have said here. Please see our point-by-point responses below.
Major comments:
- The investigation of only 2 loci needs to be explained more. I do not understand why the analysis was restricted to just two loci. In the current manuscript, this decision seems to be arbitrary. Surely, it would be more comprehensive to assess all assessed genetic variants. Indeed, hypothesis-free (i.e. not candidate gene) studies have proven to be more robust to detecting true positives (i.e. results that can be replicated). I have strong reservations about the choice of selecting only 2 loci without much of an explanation, and I have concerns that the results deemed significant by the authors are just borderline significant by chance. In addition, I am assuming the authors used p<0.05 to denote significant, and that multiple testing was not accounted for. Again, this setting is not suitable to reliably detect true associations.
The two loci and panel of genetic markers tested were specifically selected based on our own interest in these loci (we have published GWS in affected sib pairs and Gordon syndrome families) and other extensive literature of the loci suggesting direct involvement in essential hypertension. In the revised manuscript, we have clarified this point and provided additional references.
We did not perform a GWAS on our subjects due to time and budgetary constraints. Yes, a GWAS is hypothesis-free and is designed to generate regions of interest that can be probed further with techniques such as discordant sib-pairs. Given the state of knowledge and our interest in these loci we think it is scientifically credible to test these loci in a hypothesis-driven way knowing that a posteriori probability of linkage may be higher than surrounding unlinked loci.
Having performed GWAS and eQTL studies ourselves, we are keenly aware of the risks of false positive findings and multiple testing, but as stated in the manuscript, our study strengths are its design, selection of sib pairs in families with EH locally with lower risk of cardiovascular disease, a genotyping strategy using individuals from the two tails of blood pressure distribution (high and low BP sibs) and validation of previous markers.
We purposefully did not correct for multiple testing as we were only seeking correlations between markers and blood pressure phenotypes in small numbers. This was not a GWAS which looked at 1000s or even 100s of SNPs/markers where multiple testing is a very real issue.
- With regard to the description of the cohort and the demographic table. It is written that gender is assessed. Can the authors confirm that indeed they mean self-reported gender as asocial construct or are they perhaps referring to genetic sex? I usually see genetic association studies adjusting models for sex rather than gender. It is essential to use the correct terminology to avoid confusion so that is why I am asking for clarification on this point.
We apologise for the sloppy terminology over gender vs sex and it has been amended in the revised manuscript text and tables.
- All sections of the manuscript (from the introduction to conclusion) were very brief and lacked sufficient detail in my opinion.
Again, as stated above, this was a small study of 24 sib pairs and we feel we provided sufficient details in the background, methodology, data/results, discussion, conclusions and references. We did not feel it was essential to provide extensive citations and focussed only on those few studies which used this strategy to corroborate our findings.
Minor comments:
- Human gene names should be italicized.
We apologise for this mistake and in the revised manuscript text all gene names have been italicised.
- When providing the mean in the text, also include the corresponding standard deviation, as what is done in the demographic table (e.g. for the mean age of the population provided in the abstract).
As requested, we have now provided the corresponding standard deviations for the means in the revised abstract and manuscript text.
- For the demographics table, I am curious as to why Birthweight is in pounds, but the other weights are in Kg. Is there a reason for this change in units?
Demographic information was collected during Clinic visits at the hospital, and self-reported birth weight was in pounds and ounces and presented per se. Whereas, height and weight were measured using standard methods in meters and kilograms, and these units were used to calculate the body mass index as kg/m3. We did not feel we had to change units for uniformity but present the data as it has been collected similar to all our previous studies. But we are happy to present birthweight in kilograms if necessary.
- Some formatting issues (e.g. in the last sentence of the conclusion there seems to be an space before the final e in the word large)
This error must have occurred whilst using the journal template, and this has been amended in the revised conclusion section.
Reviewer 2 Report
A clear and eloquent study employing an extreme discordant sib pair strategy to analyse markers for blood pressure, a complex polygenetic and environmental condition. The study is novel, and will be of interest to a wide readership (genetics), as it employs EDSPs to interrogate this complex biological puzzle. The hypothesis is clearly stated and founded on published literature and studies. The approach is a pioneering one for this clinical presentation, and the finding will have tangible and translatable implications for future studies. The data is well interrogated, interpreted and presented. The manuscript itself is well written in a clear and concise manner.
One question that may pertain to this study is the role of environmental architecture (Note Bene - diet) as opposed to genetic architecture. If possible, the authors could briefly describe this in the manuscript, but one would imagine, that being fro the same familial unit, these compounding factors would be negligible. One possibility for future studies would be to employ biomarker specific arrays (e.g. PLA panels from OLINK) to support the genetic marker studies, to provide a functional background to support genotypic findings.
Overall, an excellent manuscript based on an eloquent study.
Author Response
We would like to respectfully pick up on what the reviewers have said here. Please see our point-by-point responses below.
A clear and eloquent study employing an extreme discordant sib pair strategy to analyse markers for blood pressure, a complex polygenetic and environmental condition. The study is novel, and will be of interest to a wide readership (genetics), as it employs EDSPs to interrogate this complex biological puzzle. The hypothesis is clearly stated and founded on published literature and studies. The approach is a pioneering one for this clinical presentation, and the finding will have tangible and translatable implications for future studies. The data is well interrogated, interpreted and presented. The manuscript itself is well written in a clear and concise manner.
One question that may pertain to this study is the role of environmental architecture (Note Bene - diet) as opposed to genetic architecture. If possible, the authors could briefly describe this in the manuscript, but one would imagine, that being from the same familial unit, these compounding factors would be negligible. One possibility for future studies would be to employ biomarker specific arrays (e.g. PLA panels from OLINK )to support the genetic marker studies, to provide a functional background to support genotypic findings.
Overall, an excellent manuscript based on an eloquent study.
We thank reviewer 2 for finding our work clear, eloquent, interesting and agree it generates wide readership in the genetics field. We also agree that it does have tangible and translatable implications for future studies and take their point on environmental / dietary factors and the OLINK arrays to support genetic marker studies.
Reviewer 3 Report
The authors investigated in untreated offspring of affected sib pairs with essential hyper tension two loci: i.) the Gordon’s syndrome (psuedohypoaldosteronism type II, PHA type 2) locus on chromosome 60 17 including WNK4 gene, and ii.) the ROMK locus on chromosome.
They found an inverse correlations between the blood pressure and several microsatellite markers suggesting a linkage on chromosome 11 and 17.
Comments:
1. How many patients have been determined WNK4 gene polymorphisms?
2. Which is the concentration of renin activity, aldosterone, potassemia, chloremia in these patients?
3. There are not data to explain a linkage on chromosome 11 and 17 in these patients.
Author Response
We would like to respectfully pick up on what the reviewers have said here. Please see our point-by-point responses below.
The authors investigated in untreated offspring of affected sib pairs with essential hypertension two loci: i.) the Gordon’s syndrome (psuedohypoaldosteronism type II, PHA type 2) locus on chromosome 60 (we assume this was a typo!) 17 including WNK4 gene, and ii.) the ROMK locus on chromosome.
They found an inverse correlations between the blood pressure and several microsatellite markers suggesting a linkage on chromosome 11 and 17.
We are disappointed that this reviewer did not focus on our stated study objective, design or findings and has made what we feel are unhelpful comments based on the serum markers, linkage and ticking boxes on the report form without reasoned explanations. We are unsure what level of improvement the reviewer is expecting from such a small study.
Comments:
- How many patients have been determined WNK4 gene polymorphisms?
We clearly stated in methods that 24 extreme sib pairs were enrolled for this study and used for genetic analysis including WNK 4 gene polymorphisms (see last sentence of the methods section before statistical analysis). The subjects were not selected because they had either a PHA2 phenotype or genotype.
- Which is the concentration of renin activity, aldosterone, potassium, chloremia in these patients?
This was a genetic study and we did not measure all the above biomarkers in our subjects with a view to pre-selecting a PHA2 phenotype (the chloride in particular would only be relevant for PHA2) as we have emphasised above. However, in the revised manuscript we analysed and provided data on the markers that are routinely measured in a clinical setting. We have now incorporated this data in the revised Table 2 and manuscript text, and as can be seen from revised Table 2 the serum markers did not differ between the two groups.
- There are not data to explain a linkage on chromosome 11 and 17 in these patients?
Our hypothesis was that these loci that contain proteins known to be involved in blood pressure (BP) regulation might contain common variants that are linked to a BP phenotype. We found evidence of linkage QED. We did not perform a GWAS linkage analysis to look at the entirety of chromosomes 11 and 17, but actually sought correlations between genetic markers in these loci and blood pressure phenotypes. We confirmed the significant correlations observed as evidence of linkage using both the Haseman-Elston quantitative trait loci regression method and SPLINK package in this sib pairs.